

# Comparing autoregulatory progressive resistance exercise and velocity-based resistance training on jump performance in college badminton athletes

Zijing Huang[1,*], Hongshen Ji[2,*], Lunxin Chen[1], Mingyang Zhang[1], Jiaxin He[2], Wenfeng Zhang[1], Xin Chen[1], Jian Sun[2], Junyi Song[3] and Duanying Li[2]

[1] Digitalized Performance Training Laboratory, Guangzhou Sport University, Guangzhou, Guangdong, China
[2] Sports Training Institute, Guangzhou Sport University, Guangzhou, Guangdong, China
[3] Graduate School, Guangzhou Sport University, Guangzhou, Guangdong, China
* These authors contributed equally to this work.

Corresponding authors
Junyi Song, 11088@gzsport.edu.cn
Duanying Li,
liduany@gzsport.edu.cn

## ABSTRACT

**Objectives:** Jumping ability has been identified as a key factor that influences the performance of badminton athletes. Autoregulatory progressive resistance exercise (APRE) and velocity-based resistance training (VBRT) are commonly used approaches to enhance muscle strength and have been shown to accurately monitor the development of explosive power to improve jumping ability. This study aims to investigate the effects of APRE and VBRT on badminton athletes' jumping ability and to provide practical insights into improving their jumping performance during competitions.

**Methods:** Upon completing familiarization and pretesting, 18 badminton athletes were included and completed the training intervention (age, 21.4 ± 1.4 years; stature, 170.1 ± 7.3 cm; body mass, 65.9 ± 12 kg); they were randomly divided into the APRE group ($n = 9$) and VBRT group ($n = 9$). Jumping performance was assessed during the countermovement jump (CMJ), squat jump (SJ), and drop jump (DJ) *via* SmartJump, with CMJ's and SJ's jump height, eccentric utilization ratio (EUR), and reactive strength index (RSI). All participants then completed a 4-week in-season resistance training intervention.

**Results:** (1) The results of the within-group indicated that only the CMJ (pre: 41.56 ± 7.84 *vs* post: 43.57 ± 7.85, $p < 0.05$) of the APRE group had significant differences, whereas the SJ, EUR, and RSI were not significantly different ($p > 0.05$). (2) The results of the intergroups revealed that all indicators had no significant differences ($p > 0.05$), but APRE had a moderate effect size on the improvement of the CMJ ($\eta^2 = 0.244$) and EUR ($\eta^2 = 0.068$) when compared with VBRT.

**Conclusions:** The results showed that, compared to VBRT, APRE can effectively improve the performance of the reactive athletes' lower limb explosive power in the CMJ in a shorter period of time. The findings indicate that APRE may be useful for coaches seeking to improve the CMJ performance of athletes in the short term.

# INTRODUCTION

Badminton is a type of racquet sport that involves various explosive movements, such as jumping, quick changes of direction, net shots, and fast arm movements (*Cabello Manrique & Gonzalez-Badillo, 2003*). It requires specific physical and physiological attributes, court performance, and technical and tactical requirements (*Faude et al., 2007*). Among these, jumping ability has been identified as a key factor that influences the performance of badminton athletes (*Ferreira, Gorski & Gajewski, 2020*). It is well understood that muscle strength plays a key role in improving and maintaining sports performance, including speed (*Chelly et al., 2009*), agility (*Spiteri et al., 2015*), and explosive strength (*Chelly et al., 2009*; *Andersen et al., 2010*), and it even contributes to the development of motor performance (*Suchomel, Nimphius & Stone, 2016*). Resistance training is recognized as an effective method for improving explosive power (*Kraemer & Ratamess, 2004*). In recent years, the autoregulatory resistance training (ART) method has been proposed and applied to strength training to overcome the shortcomings of percentage-based resistance training (PBT) (*Shattock & Tee, 2020*; *Graham & Cleather, 2021*). The primary objective of this method is to monitor and assess whether the exercise load is suitable for the athlete's motor condition in real time, while also adjusting the training load (such as volume and intensity) to ensure that the athlete trains with an appropriate load that matches their immediate condition, thereby achieving better results and reducing the risk of fatigue fatigue (*Flanagan & Jovanovic, 2014*).

Autoregulatory progressive resistance exercise (APRE), rating of perceived exertion (RPE), and velocity-based resistance training (VBRT) are three common methods of ART (*Zhang et al., 2021*). The three methods are effective in controlling the intensity of resistance training (*Zhang et al., 2021*). VBRT is an emerging and popular method of monitoring and designing prescription training that uses various advanced speed measurement devices during strength training, purposefully tracks the speed of moving loads, and provides feedback (*Zhang et al., 2022*). APRE is a daily adjustable progressive resistance exercise (*Weber, 2015*) similar to nonlinear programming resistance training, where the number of repetitions performed until exhaustion in the last two sets are used to monitor and adjust the athlete's training load for both the current and subsequent training sessions (*Ghobadi et al., 2022*). Among them, APRE's 6RM program (*Mann, 2011*) and VBRT's 10% velocity loss (*Pareja-Blanco et al., 2020*) are commonly used to develop athletes' explosive power and have been proven to accurately monitor the development of muscle explosive power.

In the field of ART, attention has been focused on comparing APRE with PBT (here, PBT refers to linear periodized percentage-based resistance training, and all studies on APRE in this article used this definition of PBT), VBRT with PBT, VBRT with RPE, and different velocity losses (*Huang et al., 2022*). However, no studies have compared APRE with VBRT, and the advantages of these two methods for developing muscle strength

**Table 1 Physical characteristics of the participants (M ± SD).**

|  | APRE (*n* = 9) | VBRT (*n* = 9) | *p* |
|---|---|---|---|
| Age (years) | 21.2 ± 1.39 | 22.1 ± 1.52 | 0.468 |
| Hight (cm) | 172.4 ± 7.18 | 168.9 ± 7.93 | 0.842 |
| Weight (kg) | 66.46 ± 10.74 | 63.99 ± 13.27 | 0.787 |

remain to be elucidated. Therefore, the purpose of this study was to compare the effects of APRE and VBRT on explosive power in a medium cycle, which could provide a practical basis for selecting a more appropriate method of autoregulatory resistance training.

# MATERIALS AND METHODS

## Experimental approach to the problem

A randomized controlled trial was conducted to investigate the effects of APRE and VBRT on the explosive power of the lower limbs in athletes. After familiarization with the program and pretesting, participants were randomly assigned to the VBRT or APRE group. Participants volunteered to participate in the study from April 2022 to June 2022. The experimental period lasted for 6 weeks, including the first and last weeks of testing, in which all participants completed two training sessions per week for 4 weeks. Testing consisted of countermovement jump (CMJ), squat jump (SJ), and drop jump (DJ) tests. All tests were performed at least 48 h before/after the most recent training session. All testing and training sessions took place at the same venue under the direct supervision of the lead investigator.

## Participants

Twenty-one participants (male = 11, female = 10) were originally recruited to take part in the research study. The inclusion criteria of the recruited athletes for the study were as follows: (1) absence of any significant physical health issues, (2) older than 18 years, (3) a minimum of 3 years of prior experience in playing badminton, and (4) absence of any injury or illness in the past 6 months. However, three participants were excluded due to injury, so only 18 badminton athletes were included and completed the training intervention (age, 21.4 ± 1.4 years; stature, 170.1 ± 7.3 cm; body mass, 65.9 ± 12 kg). The participants were randomly distributed into two groups: VBRT (*n* = 9) and APRE (*n* = 9). The participants' characteristics are presented in Table 1. There was no statistically significant difference between the two groups in terms of baseline (*p* > 0.05), and the grouping was reasonable. All of the included athletes were free of injury and sleep disorders and were nonsmokers. They volunteered to participate in this study and provided signed informed consent after being informed of the testing procedure and potential risks. This study was approved by the Ethics Committee for Human Experiments (approval number 2022LCLL-38).

## Procedures

Participants completed jumping performance testing in 1 day, including the CMJ, SJ, and DJ. Before all testing and training sessions, participants were supervised during a standardized warm-up, consisting of 5 min of jogging and dynamic stretching. After the completion of the final resistance training, testing for outcome measures was repeated after 48 h of recovery.

## Outcome measures

CMJ, SJ, and DJ tests were administered indoors using the SmartJump wireless portable jump test mat, which consists of a wireless mobile device terminal and a SmartJump vertical jump mat. The jumping test mat analyzes the jumping motion by pressure sensing. All the tests required participants to place their arms around their waist and not bend their hips and knees during the lift-off process. For the CMJ, participants kept their torso as immobile as possible while simply completing a coherent and rapid squat jump to the maximum height. For the SJ, participants listened to the experimenter's command; after the participants hear "squat," they perform a half squat; the experimenter says "1, 2," and then "jump," and the participants quickly jump to their maximum height. For the DJ, participants stepped off a box (height, 30 cm) with their preferred leg, landed on the floor with both feet, and immediately jumped as high as possible. Participants were instructed to jump to the highest height as fast as possible. The CMJ, SJ, and DJ were performed three times with 20-s rests between each jump, with the highest jump (in centimeters), eccentric utilization ratio (CMJ's jump height (in centimeters)/SJ's jump height (in centimeters)), and reactive strength index (jump height (in meters)/contact time (in seconds)) used for analysis.

## Training routine

Resistance training started at 3:00 PM on Mondays, Wednesdays, and Fridays and ended by 5:00 PM. This study used GymAware, a jump box, a barbell, and other equipment. Each resistance training session began with a standardized warm-up followed by two sets of 10 free-weight back squat repetitions, separated by 2 to 3 min of active rest periods. The APRE group performed back squats with a 6RM session based on their baseline one-repetition maximum (1RM), whereas the VBRT group performed back squats with a modifiable load based on a target velocity threshold established from the standardized load–velocity relationships.

*Autoregulatory progressive resistance exercise.* Participants in the APRE group performed with a set number of repetitions at a certain percentage of the 6RM based on Delorme's PRE program. The 6RM resistance training protocol consisted of four sets. Set 1 used 50% 6RM for 10 squats, and Set 2 used 75% 6RM for six squats. In Set 3, participants were required to squat with 100% 6RM load until failure, and then the squat weight of Set 4 was determined according to the load adjustment table (refer to Table 2). In Set 4, participants were required to perform repetitions as hard as possible until exhaustion, and the number of repetitions was used to specify the starting load for the next training session (*Verkhoshansky & Siff, 2009*).

**Table 2 APRE protocol for 6RM and Set 4 adjustment.**

| Repetitions | Intensity (% of 6RM) |
|---|---|
| APRE protocol for 6RM | |
| 10× | 50% |
| 6× | 75% |
| Maximum | 6RM |
| Maximum | Adjusted weight |
| Repetitions for set 3<br>  6RM routine adjustment | Set 4 adjustment |
| 0–2 | −2.5 to 5 kg |
| 3–4 | −0 to 2.5 kg |
| 5–7 | keep |
| 8–12 | +2.5 to 5 kg |
| 13+ | +5 to 10 kg |

Note:
APRE, autoregulatory progressive resistance exercise; 6RM, 6 repetition maximum.

*Velocity-based resistance training.* Participants in the VBRT group performed with a load that corresponded to mean concentric velocity (MCV) at 80% 1RM established from a standardized load–velocity relationship. MCV monitoring was used to dictate changes in load lifted and the number of repetitions completed on a real-time, set-by-set basis. Velocity stops were integrated into each set at 10% below the target velocity of each specific zone (*Pareja-Blanco et al., 2017*). Thereafter, if the maximum MCV in a set of repetitions was ±0.06 m/s outside of the target movement velocity, the barbell load was then adjusted by ±5% 1RM for the subsequent set (*Orange et al., 2019*).

## STATISTICAL ANALYSES

Data analysis was completed using Jamovi 2.3.26. Mean and standard deviation (SD) values were calculated using standard statistical methods. The normality of all variables was tested using the Shapiro–Wilk test procedure. Levene's test was used to determine the homogeneity of variance. Subsequently, repeated measures analysis of variance was performed. The differences between the groups before and after training and the differences in change scores between the two groups before and after training were compared. The Bonferroni adjustment was performed to determine the $p$ value of the comparisons, and $p < 0.05$ was considered statistically significant. The partial squares eta ($\eta^2$) is a measure of effect size for between-group differences in intervention effects that were calculated and considered small ($0.01 \leq \eta^2 \leq 0.06$), moderate ($0.06 \leq \eta^2 < 0.14$), or large ($\eta^2 \geq 0.14$) (*Muller, 1989*).

## RESULTS

All data are expressed as means ± SD. The subject's characteristics including age, weight, and height are listed in Table 1. There were no significant differences between groups for any of the participants' characteristics and performance outcomes at the baseline ($p > 0.05$). Likewise, repetitions and total training volume had the same results.

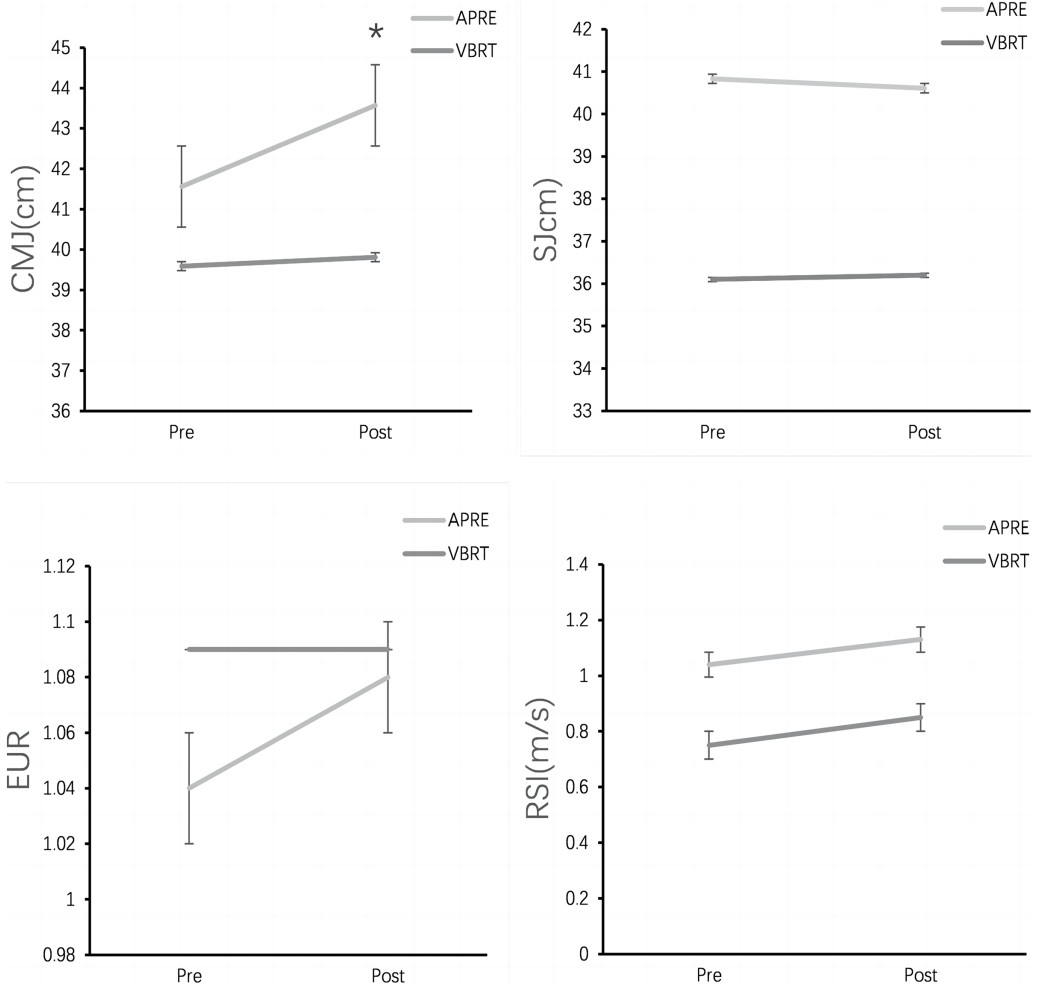

**Figure 1 Intra-group and inter-group change in CMJ, SJ, EUR, and RSI from 4 week for APRE and VBRT groups.** All data are presented as Mean ± SD. *Significant differences between pre- and post- for APRE or VBRT at the $p < 0.05$.

After 4 weeks of training, only the APRE group demonstrated greater improvement in the CMJ (pre: 41.56 ± 7.84 *vs* post: 43.57 ± 7.85; $p = 0.04$) when compared with the baseline. Pre- to post-training changes in the CMJ, SJ, eccentric utilization ratio (EUR), and reactive strength index (RSI) were compared between the APRE- and VBRT-trained groups. There were no significant differences between groups for all measures ($p > 0.05$) (Fig. 1). The effect size indicates that APRE is better than VBRT in the CMJ ($\eta^2 = 0.244$) and EUR ($\eta^2 = 0.068$), relatively (Table 3).

## DISCUSSION

In this study, APRE was found to be effective in improving the CMJ's jump height of college badminton athletes over a 4-week period. However, the between-group comparison between APRE and VBRT indicated no significant difference in the improvement of the jumping ability of the athletes. Despite this finding, considering the large and moderate effect sizes (*Muller, 1989*) of APRE on improving the CMJ and EUR of athletes compared

**Table 3 Changes in the variables of CMJ, SJ, EUR, and RSI between pre- and post-test after training in each of groups.**

| Variables | APRE (n = 9) | | | VBRT (n = 9) | | | Effect sizes (η²) |
|---|---|---|---|---|---|---|---|
| | Pre | Post | p value | Pre | Post | p value | |
| CMJ (cm) | 41.56 ± 7.84 | 43.57 ± 7.85 | 0.04 | 39.59 ± 7.97 | 39.81 ± 7.08 | 1 | 0.244 |
| SJ (cm) | 40.83 ± 7.69 | 40.61 ± 8.37 | 1 | 36.1 ± 7.35 | 36.2 ± 6.33 | 1 | 0.003 |
| EUR | 1.04 ± 0.06 | 1.08 ± 0.04 | 0.986 | 1.09 ± 0.07 | 1.09 ± 0.05 | 1 | 0.068 |
| RSI | 1.04 ± 0.27 | 1.04 ± 0.36 | 1 | 0.75 ± 0.21 | 0.85 ± 0.26 | 1 | 0.000 |

with VBRT, a previous study suggested that APRE is more effective than PBT for short-term gains (*Mann, 2011*). It can be concluded that APRE has a positive effect on improving the pre-competition jumping ability of college badminton athletes.

The three most popular methods for assessing vertical jump are the CMJ, SJ, and DJ (*Kozinc, Plesa & Sarabon, 2021*), and it is well-known that participants can achieve higher jump heights and greater output power in the CMJ compared with the SJ (*Bobbert et al., 1996*). A study suggested that 4 weeks of PBT did not significantly improve the CMJ and SJ of athletes (*Kovacevic et al., 2022*). However, the results of the present study were different; after 4 weeks of training, the participants' CMJ in the APRE group was significantly improved. Mann also highlighted that APRE is more applicable to athletes than PBT (*Mann et al., 2010*). PBT, VBRT, and APRE employ different methods for setting the intensity and volume of resistance load during a resistance training session. PBT prescribes the weight, the number of repetitions, and sets for each squat (*Zhang et al., 2022*); VBRT sets a speed range for the squats, and training is immediately stopped once the speed falls out of the range (*Zhang et al., 2022*); APRE aims to maximize the number of repetitions (*Mann et al., 2010*). Specifically, APRE sets the overall number of sets and the number of repetitions for the first two sets, whereas the third and fourth sets require the subject to reach exhaustion, but the maximum number of repetitions varies with the subject's exercise status (*Mann et al., 2010*). The third and fourth sets of APRE required participants to perform repetitions until failure, and the rationale for performing resistance exercises until failure is to maximize motor unit recruitment (*Willardson, 2007*). Type IIa muscle fibers are maximally mobilized during force training under moderate-to-high intensity loads. Because of its greater aerobic metabolic capacity and fatigue resistance (*Castro et al., 1998*; *Hather et al., 1991*), fatigue resistance is more conducive to the development of muscle strength and muscle hypertrophy (*Mang, Rasinski & Kravitz, 2022*). As the improvement of the CMJ in adult athletes largely depends on the increase in individual muscle strength, and the training mechanism of APRE is more conducive to inducing the development of maximum strength (*Rebelo et al., 2022*), it is plausible that APRE is better than VBRT for improving the CMJ after 4 weeks of training.

The ratio of the CMJ to the SJ is known as the EUR (*Mcguigan et al., 2006*) and is often used to examine the effect of reversal on generating greater jump height and the effect of the neuromuscular system on rapid force generation (*Hasson et al., 2004*). After 4 weeks of

resistance training, the EUR of VBRT remained stable, and its CMJ and SJ did not change significantly before and after the intervention training. The EUR of APRE was improved to 1.08 (1.1 was the best optimum) (*Balsom, 1994*), whereas its SJ did not significantly change after the training (pre: 40.83 ± 7.69, post: 40.61 ± 8.37). However, the CMJ (pre: 41.56 ± 7.84, post: 43.57 ± 7.85) indicated significant improvement after the training. According to the EUR standard calculation formula, it was found that the improvement in APRE's EUR was mainly caused by the increase in the CMJ. The exhaustion training of the last two groups in the APRE training mechanism appears to be an important factor in improving EUR to a high level. As mentioned earlier, exhaustion training in APRE can maximize the recruitment of motor units to increase muscle strength and improve the CMJ (*Rebelo et al., 2022*). Therefore, the improvement in EUR can be attributed to the increase in the recruitment ability of motor units. The improvement of EUR indicates an increase in the storage and utilization of elastic energy in the stretch-shortening cycle, which means that athletes can better perform muscle pre-activation, the stretch reflex, and the release of stored passive elastic energy in the muscle tendinous tissue (*Groeber, Stafilidis & Baca, 2021*). Although there was no significant difference in EUR between the two groups in the pre-test, the EUR of VBRT was higher than that of APRE at 0.05, which was close to the optimal value of 1.1. Therefore, this result only indicates that APRE can increase athletes' EUR, but the effect of VBRT on EUR is unclear. Small optimization space may lead to non-significant study findings, reducing the research's impact.

*Young (1995)* proposed the RSI as a measure of the ability to quickly transition from eccentric to concentric muscle contractions during jumping. The RSI is calculated as the jump height during a DJ divided by the contact time (*Louder, Thompson & Bressel, 2021*). Improvement of the RSI can be achieved by increasing jump height and decreasing contact time. The meta-analysis of *Rebelo et al. (2022)* found that both long-term (*Keiner et al., 2018*) and short-term (*Orange et al., 2019*; *Murton, Eager & Drury, 2023*) resistance training can increase the RSI in adolescents, but the effect on trained adult athletes is smaller. Additionally, 4 weeks of both high- and low-intensity resistance training failed to significantly improve explosive strength in adult athletes (*Argus et al., 2012*). Resistance training typically involves moderate-to-high loads (*Cormie et al., 2007*), which results in relatively slow movement speeds that may not be conducive to rapid force generation. The improvement in the RSI in adolescents is attributed to two factors: (1) an increase in muscle strength through resistance training leading to an increase in jump height while contact time remains constant (*Young, 1995*) and (2) an increase in activated muscle units, where resistance training can activate 10% more muscle units in adolescents than in adults (*Lloyd et al., 2012*). However, for adult athletes, it is challenging to significantly improve the RSI through resistance training once muscle strength has reached a certain level, but reducing contact time may improve the RSI. Research has suggested that post-activation potentiation can have an acute effect on the RSI in adult athletes. After heavy squats at 93% of one-repetition maximum, the RSI was slightly improved but not significantly, contact time decreased significantly (7.8%), and leg stiffness improved significantly (10.9%) possibly due to increased neural-muscular activation, allowing athletes to better control the stiffness of their leg muscles and thereby reduce contact time (*Comyns et al., 2007*).

Huang et al. (2023), *PeerJ*, DOI 10.7717/peerj.15877

Similarly, plyometric training can improve the RSI in adult athletes by reducing contact time during jumping (0.84, 95% CI [0.37–1.32]) (*Rebelo et al., 2022*). Therefore, resistance training can improve the RSI in adolescents by increasing muscle strength, but improving the RSI in adult athletes through resistance training is challenging. Post-activation potentiation and fast stretch-shortening combined training can improve the RSI in adult athletes by reducing contact time.

In this study, the participants were college badminton athletes, and the intervention involved a 4-week resistance training program using the autoregulation method. The results showed a significant improvement in jumping ability for the athletes in the APRE group. However, this study has several limitations that should be mentioned: (a) a control group was not used to compare the effect of the current experimental protocols, (b) the number of participants was not sufficient, (c) a relatively short training intervention was used, and (d) the external validity of this study may be limited to athletes in sports similar to badminton. The ART allows athletes to adjust the intensity of their training based on their own abilities and fatigue levels, which may improve performance and reduce the risk of injury. Although the results of this study may be limited to specific contexts of sample selection and intervention duration, they may still be a useful tool for coaches and athletes to design more effective resistance training programs. Future research directions may include investigating the long-term effects of ART on athletes' performance, as well as studying its effects on other athletic performance measures beyond jumping ability. Additionally, the effectiveness of the ART can be studied in athletes from different sports or at different levels of competition.

## CONCLUSION

Improving the jumping ability of badminton athletes is crucial for achieving excellent athletic performance and competition results. Therefore, coaches need to understand methods for enhancing the jumping ability of athletes during the season. This study found that using APRE can effectively improve the CMJ performance of badminton athletes in a shorter period and that APRE is superior to VBRT in terms of CMJ performance and EUR testing. From a practical perspective, APRE was associated with greater power and neuroadaptation, as well as similar repetitions and total training volume, throughout a 4-week training cycle when compared with VBRT. Therefore, APRE may assist coaches in enhancing the CMJ performance of athletes in the short term, whereas improvements in the SJ performance, EUR, and RSI may require longer-term training intervention to manifest.

## ACKNOWLEDGEMENTS

We would like to thank the researchers and study participants for their contributions.

### Funding

The authors received no funding for this work.

## Competing Interests

The authors declare that they have no competing interests.

## Author Contributions

- Zijing Huang conceived and designed the experiments, prepared figures and/or tables, and approved the final draft.
- Hongshen Ji conceived and designed the experiments, prepared figures and/or tables, and approved the final draft.
- Lunxin Chen conceived and designed the experiments, prepared figures and/or tables, and approved the final draft.
- Mingyang Zhang conceived and designed the experiments, prepared figures and/or tables, and approved the final draft.
- Jiaxin He performed the experiments, prepared figures and/or tables, and approved the final draft.
- Wenfeng Zhang performed the experiments, prepared figures and/or tables, and approved the final draft.
- Xin Chen performed the experiments, prepared figures and/or tables, and approved the final draft.
- Jian Sun analyzed the data, authored or reviewed drafts of the article, and approved the final draft.
- Junyi Song analyzed the data, authored or reviewed drafts of the article, and approved the final draft.
- Duanying Li analyzed the data, authored or reviewed drafts of the article, and approved the final draft.

## Human Ethics

The following information was supplied relating to ethical approvals (*i.e.*, approving body and any reference numbers):

This study was approved by the Ethics Committee for Human Experiments (Approval number: 2022LCLL-38).

## Data Availability

Raw data are available in the Supplemental Files.

## Supplemental Information

Supplemental information for this article can be found online at http://dx.doi.org/10.7717/peerj.15877#supplemental-information.

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
