# Peer review of "Comparing autoregulatory progressive resistance exercise and velocity-based resistance training on jump performance in college badminton athletes"

_PeerJ, doi:10.7717/peerj.15877_

## Round 0.1 · original submission · Major Revisions

Dear Authors,

The reviewers and I have completed our evaluation of your manuscript and recommend a major revision before re-submission.

Please review the comments and resubmit your revised manuscript.

·

Basic reporting

I would thanks the authors for the effort and for the best quality of the manuscript. However, there is some mistakes to correct them.
General comments:
- The authors need to work with a fluent English speaker/writer to correct grammatical and punctuation errors throughout the manuscript.

Abstract
- The aim of the study is not clear
- Add the materials used in this part
- Conclusion need to be reformulated.
Introduction
- L37- L40: This part is not clear
- L44- L46: Add reference
- L48- L50: Add reference
- L48- L54: There’s no consistency between paragraphs
- The problematic is not clear

Experimental design

Methods
- Experimental approach need to be more clear for the reader
- Why you choose participants of both sexes
- I think that the number of participants is small
- Inclusion and exclusion criteria need to be more clearer

Validity of the findings

Results
- Please improve the image quality of figure1
Discussion
- L160-L163: Add reference
- L164-L168: This paragraph is not clear
- L169-L176: This result is not well discussed
- Discuss limitations of the study, taking into account sources of potential bias or imprecision. Discuss both direction and magnitude of any potential bias
- Discuss the generalisability (external validity) of the study results
- Discuss the practical implications and future research

Conclusions

- Thanks to reformulate conclusions part

Reviewer 2 ·

Basic reporting

The article is written in good and understandable English language. I do have a single critique since I have noticed that there are some acronyms, which are missing their extensive form the first time they appear: lines 48, 120.
In addition, I suppose there is a typo at line 115.

Experimental design

The experimental design and protocol are well-defined and seem coherent with the research question.
However, I would suggest changing the spot of the males/females count since it does not belong to its current position: the sentence at line 74 starts as "twenty-one subjects.." while the sum of males/females count in parenthesis is 18. Therefore, you could rephrase the sentence and maybe move the parenthesis to a spot where you describe the number of participants left.

Validity of the findings

Ranganathan et al., (Perspect Clin Res. 2016 Apr-Jun;7(2):106-7. doi: 10.4103/2229-3485.179436.) have described the problems of multiple statistical testing and why it should be avoided. In lines 129-131, you describe the different tests you used to compare the groups. However, no direct comparisons between groups were performed in the statistical analysis. You performed different t-tests to compare groups at baseline and to compare pre- vs. post-training outcomes in the same group, while you should perform the ANOVA statistics (+ post-hoc tests) instead.
Moreover, it is still unclear how you compared groups. In addition, you did not describe g-value at line 141.
Using Cohen's d to compare different interventions is not a valid and reliable way to determine whether one intervention is better than the other, and it cannot substitute a direct statistical comparison performed with an ANOVA.

·

Basic reporting

The paper approaches an interesting theme and has been well-written and well-organized. The introduction brings an explanation of basic concepts and the literature concerning the study question.

Experimental design

I suggest removing the randomized controlled trial from the title, considering that the study does not have a control group.

Validity of the findings

I suggest writing the discussion mainly from the second paragraph where the authors bring the concepts well written but do not relate these concepts with the findings of the study. As well as describing more clearly the cited studies presenting results and comparing with the findings in the present study

---

## Round 0.2 · Major Revisions

Dear Authors,

The reviewers and I have completed our evaluation of your manuscript and recommend a major revision before re-submission.

Please review the comments and resubmit your revised manuscript.

·

Basic reporting

- The authors need to work with a fluent English speaker/writer to correct grammatical and punctuation errors throughout the manuscript.

- Title of paper need to be reformulated

Abstract
- The aim of the study is not clear
- Conclusion need to be reorganized
Introduction
- L42-L43 : Add reference
- L44-L47: This paragraph is not clear
- L52-L56: Reformulate this paragraph
- The importance of Autoregulatory progressive resistance exercise (APRE) must be more clear.
- The problematic is not clear

Experimental design

- Why you choose participants from both genders? It can affect results or not ?
- Thanks to add the period of the season

Validity of the findings

Results
- Figure 1 not very clear.
Discussion
- L176-L179: This paragraph is not clear
- L200-L202: Add reference
- L205-L208: Reformulate this paragraph
- Discuss limitations of the study, taking into account sources of potential bias or imprecision. Discuss both direction and magnitude of any potential bias
- Discuss the generalisability (external validity) of the study results
- Discuss the practical implications and future research

Reviewer 2 ·

Basic reporting

The article is written in good and understandable English language.

Experimental design

the experimental design is coherent with the aim of the research.

Validity of the findings

The authors improved this section.

Additional comments

The authors have improved their manuscript according to the given suggestions.

·

Basic reporting

The authors answered all of my comments, and they made appropriate changes in the manuscript. Therefore I recommend this manuscript be published,

Experimental design

The authors answered all of my comments, and they made appropriate changes in the manuscript. Therefore I recommend this manuscript be published,

Validity of the findings

The authors answered all of my comments, and they made appropriate changes in the manuscript. Therefore I recommend this manuscript be published,

Additional comments

The authors answered all of my comments, and they made appropriate changes in the manuscript. Therefore I recommend this manuscript be published,

---

## Round 0.3 · accepted · Accept

Your manuscript has been accepted for publication. Congratulations!

·

Basic reporting

I think that the manuscript is well corrected and now is suitable for publication.

Experimental design

This part is well corrected

Validity of the findings

This part is well corrected

Reviewer 2 ·

Basic reporting

No comment

Experimental design

No comment

Validity of the findings

No comment

Additional comments

No comment